# Synthesis of Room Temperature Curable Polymer Binder Mixed with Polymethyl Methacrylate and Urethane Acrylate for High-Strength and Improved Transparency

**DOI:** 10.3390/polym16101418

**Published:** 2024-05-16

**Authors:** Ju-Hong Lee, Won-Bin Lim, Jin-Gyu Min, Jae-Ryong Lee, Ju-Won Kim, Ji-Hong Bae, Pil-Ho Huh

**Affiliations:** Department of Polymer Science and Engineering, Pusan National University, Busan 46241, Republic of Korea; dlwnghd15@pusan.ac.kr (J.-H.L.); wblim@pusan.ac.kr (W.-B.L.); jgmin@pusan.ac.kr (J.-G.M.); leejr1216@pusan.ac.kr (J.-R.L.); kjw8618@pusan.ac.kr (J.-W.K.)

**Keywords:** urethane acrylate, polymer binder, photopolymer, curable polymer

## Abstract

Urethane acrylate (UA) was synthesized from various di-polyols, such as poly(tetrahydrofuran) (PTMG, Mn = 1000), poly(ethylene glycol) (PEG, Mn = 1000), and poly(propylene glycol) (PPG, Mn = 1000), for use as a polymer binder for paint. Polymethyl methacrylate (PMMA) and UA were blended to form an acrylic resin with high transmittance and stress-strain curve. When PMMA was blended with UA, a network structure was formed due to physical entanglement between the two polymers, increasing the mechanical properties. UA was synthesized by forming a prepolymer using di-polyol and hexamethylene diisocyanate, which were chain structure monomers, and capping them with 2-hydroxyethyl methacrylate to provide an acryl group. Fourier transform infrared spectroscopy was used to observe the changes in functional groups, and gel permeation chromatography was used to confirm that the three series showed similar molecular weight and PDI values. The yellowing phenomenon that appears mainly in the curing reaction of the polymer binder was solved, and the mechanical properties according to the effects of the polyol used in the main chain were compared. The content of the blended UA was quantified using ultravioletvisible spectroscopy at a wavelength of 370 nm based on 5, 10, 15, and 20 wt%, and the shear strength and tensile strength were evaluated using specimens in a suitable mode. The ratio for producing the polymer binder was optimized. The mechanical properties of the polymer binder with 5–10 wt% UA were improved in all series.

## 1. Introduction

External stimulus response technology using smart materials is emerging as an attractive research field. To reduce automobile accidents that occur at night, there is a need for research on smart materials that can be applied to road surfaces. External stimulus response technology is an important skill for securing driving stability for traffic weaknesses and drivers, and it is a technology that makes it possible to recognize real time road information [1,2,3,4,5,6,7,8]. Photo stimulation-sensitive technology can be used in various fields by mixing a photo-stimulated material and various polymer resins with the property of emitting light. A representative photo stimulation-sensitive (sunlight, street lighting, car headlights, etc.) technology is a luminescent paint binder used in road and automobile lines. The polymer binders used in luminescent paints show high efficiency and stability in dark spaces where there is no light. Continuously illuminated painted lanes create a safe traffic environment that provides information to drivers and prevents accidents. Therefore, polymer binder manufacturing technology that mixes with phosphorescent pigments to safely protect the pigments and can be used in the next lane is a technology that is attracting attention. The synthesis of binders that can protect pigments has become an important research goal. A photo-stimulated binder that exhibits excellent luminous properties requires the following mechanical properties: transmittance that does not reduce the luminous properties of the phosphorescent paint, adhesion to the road surface, weather resistance to withstand weather changes, and toughness to withstand external shocks [9,10,11,12,13,14,15,16,17,18,19,20,21,22,23]. Therefore, polymethyl methacrylate (PMMA) is a representative polymer used in paint manufacturing, but it has the disadvantage of inappropriate physical properties when used independently. To solve these problems, this study used PMMA and a composite series added with various acrylates. Among numerous acrylates, UA has excellent mechanical properties such as weather resistance, abrasion resistance, and alkali resistance, and structurally exhibits a network or linear structure depending on the type of polyol (diol, triol, etc.). Urethane has the advantage of varying mechanical properties (impact resistance, friction resistance, wear resistance, etc.) depending on the type of polyol and isocyanate. The use of diol and isocyanate with a linear structure can be applied to paints with excellent performance. Therefore, the UA synthesized in this study exhibits improved toughness and optical properties when blended with PMMA to produce a binder. In addition, polymer binders for road surfaces must exhibit high abrasion resistance to withstand external forces such as vehicle weight and speed, and UA is suitable because it provides the properties necessary for the purpose of use in paint. To successfully blend UA with PMMA, it is essential to synthesize them as oligomers. Their high molecular weight has the disadvantage of poor processability due to viscosity. Also, insufficient urethane bonding will cause phase separation from PMMA, so it is essential to take care. Therefore, in this study, a blend process of UA and PMMA was performed after synthesizing oligomeric UA [24,25,26,27,28,29,30,31]. PMMA is synthesized through a copolymerization process using a butyl acrylate monomer and a methyl methacrylate (MMA) monomer and has excellent optical properties, high weather resistance, and adhesive strength [32]. Urethane prepolymer is formed through a urethane bond by reacting the hydroxyl group of polyols with an NCO group of isocyanates. UA was synthesized by adding an acrylic group and capping the ends, resulting in high chemical resistance and tensile strength values. Representative polyols for synthesizing UA include polyester and polyether polyols, as well as PPG, PEG, and PTMG. Polyether polyols with the same molecular weight were used in the range of this study. Various types of isocyanates, such as methylene diphenyl diisocyanate (MDI) and toluene diisocyanate (TDI), which are used industrially, as well as HDI and 4,4′-diisocyanatodicyclohexylmethane (H_12_MDI), were used. Urethane prepolymers were synthesized using chain structured HDI to maximize the optical properties MDI and TDI with phenyl groups were not used because of the yellowing phenomenon. As the acrylate, 2-HEMA, the most widely applied industrially, was used to react with the NCO group at the end of the urethane prepolymer [33,34,35,36,37,38,39].

Three series of UA were synthesized to observe changes in the physical properties of UA according to polyols. By controlling the added catalyst, reaction time, and temperature, UA having a similar molecular weight was prepared and blended with PMMA. Curing of acrylates currently used industrially proceeds with crosslinking using heat or light. In this study, however, polymeric binders made by blending PMMA and UA can crosslink only with the reaction heat at room temperature [40,41,42,43]. Benzoyl peroxide (BPO) was used as an initiator for the curing reaction at room temperature, and N,N-bis(2-hydroxyethyl)-*p*-toluidine (PTE) was added as a catalyst to initiate the reaction of BPO. In this study, three series of UA according to the type of polyol were synthesized, and their mechanical properties were evaluated after blending with PMMA. As a result, the highest permeability, shear strength, and tensile strength were exhibited at a UA content of 5 to 10 wt%. An optimized polymer binder was prepared by adjusting the PMMA to UA ratio while the content of the initiator and reaction promoting catalyst was fixed. In future studies, it will be mixed with luminescent paint and used as a promising photo-stimulation smart, sensitive material [44,45,46,47,48,49,50,51,52,53,54,55,56,57,58].

## 2. Material and Methods

The urethane binder was synthesized using polypropylene glycol (PPG, Mn = 1000 g/mol, Merck KGaA, Darmstadt, Germany), polyethylene glycol (PEG, Mn = 1000 g/mol, Merck KGaA, Darmstadt, Germany), polytetrahydrofuran (PTMG, Mn = 1000 g/mol, Merck KGaA, Darmstadt, Germany) as the polyol of UA, and hexamethylene diisocyanate (HDI, Mn = 168.2 g/mol, Tokyo Chemical Industry Co., Ltd., Tokyo, Japan) as the isocyanate of the main chain. PMMA (Mn = 28,000, Jungseok Chemical Co., Ltd., Jeonju, Republic of Korea) was used to form a polymer binder by complexing with UA. MMA (Mn = 100.121 g/mol, Merck KGaA, Darmstadt, Germany) was used as a diluent to control the viscosity of the synthesized UA. Dibutyltin dilaurate (DBTDL, Mn = 631.56 g/mol, Merck KGaA, Darmstadt, Germany) was used as an organometallic catalyst to promote the urethane reaction between the –OH group of the polyol and the isocyanate –NCO group. N,N-bis(2-hydroxyethyl)-*p*-toluidine (PTE, Mn = 195.26 g/mol, Jungseok Chemical Co., Ltd., Jeonju, Republic of Korea) and BPO (Mn = 242.23 g/mol, Jungseok Chemical Co., Ltd., Jeonju, Republic of Korea) were used as catalysts and initiators to cure the polymer binder blended with PMMA and UA, respectively.

## 3. Results and Discussion

In the case of synthesized UA, the changes in mechanical properties according to the type of polyether polyol with the same molecular weight were compared. PPG/PEG/PTMG and HDI were added at a 1:2.2 molar ratio to a 250 mL four-necked flask. The urethane prepolymer manufacturing process was set to 50–60 °C under the condition of adding 0.1 wt% tin catalyst and stirred at 100 rpm using a mechanical stirrer. The synthesis process was analyzed by Fourier transform infrared (FT-IR) spectroscopy and shown in Figure 1. The OH stretching vibration around 3700 cm^−1^ and the NCO peak around 2250 cm^−1^ decreased because of the reaction between polyol and isocyanate, and a new urethane bond reaction was generated at 1725 cm^−1^, as shown in (a), (b). UA was synthesized by reacting the NCO group at the terminal of the urethane prepolymer with the OH group of acrylates. In the case of 2-HEMA with an acryl group, a 2.2 molar ratio was added to react and remove the isocyanate. The C=O peak at 1725 cm^−1^ and C=C peak at 810 cm^−1^ generated by the addition of an acryl group were confirmed, as shown in (c), (d). High viscosity UA was difficult to blend with PMMA. Therefore, the reaction temperature was lowered to 30 °C, and 30 wt% of MMA was added as a diluent and mixed. 

Figure 1 shows the synthesis of UA through three polyols and the curing mechanism after blending with PMMA. Three series of polyols (PPG, PEG, and PTMG) were reacted with HDI to form urethane prepolymers, respectively. The isocyanate group of the prepolymer reacted with additionally added 2-HEMA to produce Di-UA with acrylate capped end groups. It was blended with the prepared PMMA to form a polymeric binder support. 2-HEMA added during the blending process helps improve the shear strength and tensile strength through intermolecular hydrogen bonding. PTE was added as a catalyst and blended for three hours in an ultrasonic homogenizer. BPO as an initiator was added to the successfully mixed polymer binder and sufficient friction energy was applied using a vortex mixer. The PTE promotes the initiation of the BPO, and as a result, the polymeric binder was cured by self-heating at room temperature. Typical initiation reactions occur by externally applied light and heat. It is important to note that no additional external energy is required for the synthesis of polymeric binders in this paper. The formulation of the polymer binder is shown in Table 1.

The molecular weight of the synthesized UA series was measured by gel permeation chromatography (GPC). Tetrahydrofuran (THF) was used as the GPC solvent for the measurement. In a 20 mL vial, 0.05 g of the synthesized UA and 1 mL of THF solvent were added and dissolved by ultrasound from a sonicator for 6 h. The residue was then filtered off using a 0.5μm pore-size filter. The filtered sample was introduced into the GPC using a 1 mL needle. A target molecular weight of Mn 7–8000 was achieved, and a polydispersity index (PDI) of 1.95–2.00 was confirmed, as shown in Figure 2 and Table 2.

Adding low molecular weight UA reduces the mechanical properties of the binder, which has the disadvantage of a weak structure due to the low molecular weight UA chains between PMMA polymer chains. By contrast, high molecular weight acrylates have problems with the curing time and rate at room temperature owing to their high viscosity, resulting in extremely low efficiency. In addition, the processability of the binder is very important for spraying on the road surface. When forming UA with a molecular weight of 10,000 or more, the viscosity due to the high molecular weight reduces workability. Therefore, UA with an appropriate molecular weight of 5–8000 for use in road paints was synthesized. Polymeric binders were prepared by varying the content ratio of UA in the synthesized oligomeric form and PMMA. To determine the difference in binder properties according to changes in acrylate content, the mechanical properties of a polymer binder blended with UA and PMMA were evaluated while fixing the type and content of the catalyst and initiator. All tests were conducted more than five times depending on the type of sample. The average value of the results excluding the maximum and minimum values is shown. The shear strength of the SUS specimens was measured in accordance with ASTM D1002 standard [59]. Apply one drop of the prepared binder to the end of the SUS specimen (width 25.4 mm, height 12.7 mm). Using another SUS specimen, overlap to the same size and confirm that curing occurs at room temperature. Curing is completed, and the measurement is performed using UTM. The shear strength according to the UA content was measured, as shown in Figure 3. UA synthesized using three different types of polyols was blended with PMMA at 5–10 wt% it showed the highest value and decreased gradually. In this experiment, it was possible to confirm the critical content of UA blended with PMMA. The polymers were physically entangled, resulting in a network structure and high shear strength. It was conducted according to the difference in the polyol type and UA content. The mixture of 5 wt% of UA using PPG as a polyol showed the highest value of 14.6 MPa, as shown in Figure 3 and Table 3. The methyl groups of the main chain were not packed in a linear structure and had a free volume. Therefore, PPG-UA was blended with PMMA because of its good compatibility. Compared to PTMG-UA, in the case of PPG-UA, relatively high results were obtained due to the short molecular unit chain length, that is, many acrylate functional groups. As a result, to obtain PMMA-based polymer binders with high mechanical properties, optimal design of the appropriate molecular chain structure length is required.

Figure 4 shows the UV-vis data of the three series of polymeric binders at a wavelength of 370 nm, which has the highest energy in the visible light region. The prepared binder for UV-vis measurement was formed to a thickness of 175 μm on a glass slide using a Baker applicator. As with other physical property tests, measurements were completed more than five times. It was confirmed as the average value of the results excluding the maximum and minimum values. The transmittance is especially important to improve the characteristics of polymer binders for road marking when mixed with phosphorescent pigments and mechanical strength. The low transmittance makes it impossible to control the luminous effect when the phosphorescent pigment and the polymer binder are mixed. PMMA is a representative polymer with high transmittance and showed similar or improved transmittance at 370 nm when 5 wt% of PPG/PEG/PTMG-UA was used. PPG-UA measured the highest light transmittance at 89.5%, as shown in Table 4. PEG and PTMG-UA showed low transmittance because of the lack of space for light to pass through due to the structural characteristics of the linear stack. On the other hand, the transmittance decreased gradually for binders in which the UA content exceeded 10 wt%. Curing the binder with the addition of diol UA resulted in lower transmittance because it turns hazy as the content increases. Compared to PTMG and PPG, as the content of PEG-based polymer binder increases, UV transmittance rapidly decreased. Structurally, the PEG chain forms a more linear structure than other comparative products. The linear structure is packed and makes it difficult for light to pass through. As a result, the polymer binder containing 5 wt% of UA showed an optimized value.

The curing process of the synthesized binder was performed in a mold manufactured according to the ASTM D638 standard and is shown in Figure 5 [60,61]. In this study, a Teflon mold was used to produce dog-bone specimens under ASTM D638 Type IV conditions because a release agent spraying process is necessary to separate the manufactured specimens when performing binder curing in an iron mold. Figure 6 presents the tensile strength of the polymer binder according to the UA content. The prepared specimen was measured at a speed of 10 mm/min using UTM. In the case of the polymer binder cured with the PMMA resin, the stress was 7.1 MPa, whereas the binder containing 5 wt% PPG-UA showed a stress of 9.7 MPa. PMMA/PEG-UA and PMMA/PTMG-UA showed 5.2 MPa and 3.5 MPa, respectively, at a 10 wt% content. Table 5 lists the stress values for each content of the UA series. PMMA/PEG-UA was expected to show high tensile strength because of the high intermolecular attraction through hydrogen bonding, but the miscibility between PEG-UA and PMMA was poor. PMMA/PTMG-UA affects the strain rather than stress because the chain length of PTMG-based UA is long. The polymer binder prepared with 5 wt% of PMMA/PPG-UA showed approximately 30% improvement in the mechanical properties compared to the polymer binder made using PMMA resin.

## 4. Conclusions

In this study, a room temperature curing type binder series was designed as the most important element technology in externally stimulated polymer binder materials, and the limitations of commercialization were overcome by improving the low mechanical properties of PMMA based binders. In the design of a binder composed of a blend of PMMA and UA, UA using various types of polyols (PPG, PEG, PTMG) was synthesized to have the same or similar molecular weight to compare its performance evaluation. The proposed UA series was synthesized to have a molecular weight of 5–8000. Low molecular weights of 2–3000 showed very low mechanical properties for application to binders, and high molecular weights of 10,000 or more were difficult to apply due to processing problems. The optimal mixing ratio was suggested through various content splits so that the designed binder shows excellent mechanical properties. The prepared polymer binder showed improvement in transmittance, shear strength, and tensile strength at UA 5–10 wt%, and an improved polymer binder could be made with structural effects using PPG-UA. PMMA based binders blended with UA using other types of polyols (PEG, PTMG) also showed slightly improved mechanical properties. In this study, the designed PMMA/UA binder can be cured at room temperature without external energy (light, heat, etc.) due to the heat generated from the reaction with the initiator (BPO) and catalyst (PTE). A binder design that can be cured at room temperature is the most important element technology in externally stimulated polymer binders. In addition, in this study, to overcome the mechanical property limitations of PMMA based binders, UA was designed using various polyols and improved physical properties were confirmed. The proposed PMMA/UA binder material could be a promising candidate for future road marking polymer binders.

## Data Availability

Data are contained within the article.

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
