# Peer review of "Synthesis of Room Temperature Curable Polymer Binder Mixed with Polymethyl Methacrylate and Urethane Acrylate for High-Strength and Improved Transparency"

_polymers, 2024, doi:10.3390/polym16101418_

Round 1

Reviewer 1 Report

Comments and Suggestions for Authors

The publication submitted for review concerns using Urethane acrylate (UA) in manufacturing acrylic resins, the primary component of acrylic paints. The prepared polymer binders showed significantly higher properties than the reference samples. The improved properties mainly refer to permeability, shear strength, and tensile strength improvements. The paper has a structure characteristic of research papers. It contains all relevant aspects. Overall, it is interesting, but the descriptions should be organised.

My comments:

- The introduction of the work could be better. Nearly 1/3 is information about what will be done and what should be done in the future.

- There needs to be a clearly defined purpose of the work.

- If one of the elements of the thesis is a synthesis, the results should be in Chapter 3, Results and Discussion.

- Chapter 3 contains information that should have been given in Chapter 2.

- More information is needed on how the physical characteristics were measured. The presentation of literature references can only be used as supplementary information.

- The number of tests performed needs to be given.

- Fig. 3 could be more readable.

- No specific conclusions are given. This is probably due to the lack of a clearly stated research goal.

Author Response

Response to Reviewer 1 Comments

1. The introduction of the work could be better. Nearly 1/3 is information about what will be done and what should be done in the future.
Our Response : 

Additional research results were written in [Chapter 3] Results and Discussion and [Chapter 4] Conclusions.
Also, We will do additional research with new materials and approaches and I promise to submit consecutive articles with improved research results. 

2. There needs to be a clearly defined purpose of the work.
Our Response : 

Additional research Introduction was written in [Chapter 1] Introduction.

External stimulus response technology using smart materials is emerging as an at-tractive research field. To reduce automobile accidents that occur at night, there is a need for research on smart materials that can be applied to road surfaces. External stimulus response technology is an important skill for securing the driving stability for the traffic weaknesses and drivers, and it is a technology that makes it possible to recognize re-al-time road information [1-8]. Photo stimulation sensitive technology can be used in var-ious fields by mixing a photo-stimulate material and various polymer resins with the property of emitting light. A representative photo stimulation sensitive (sunlight, street lighting, car headlights, etc.) technology is a luminescent paint binder used in road and automobile lines. The polymer binders used in luminescent paints show high efficiency and stability in dark spaces where there is no light. Continuously illuminated painted lanes create a safe traffic environment that provides information to drivers and prevents accidents. Therefore, polymer binder manufacturing technology that mixes with phos-phorescent pigments to safely protect the pigments and can be used in the next lane is a technology that is attracting attention. The synthesis of binders that can protect pigments has become an important research goal.

3. If one of the elements of the thesis is a synthesis, the results should be in Chapter 3, Results and Discussion.
Our Response : 

In response to your comments, [Chapter 3] Results and Discussion has been revised as a whole.

4. Chapter 3 contains information that should have been given in Chapter 2.
Our Response : 

In response to your opinion, we have revised [Chapter 2] Materials and Methods and [Chapter 3] Results and Discussion separately.

5. More information is needed on how the physical characteristics were measured. The presentation of literature references can only be used as supplementary information.
6. The number of tests performed needs to be given.
Our Response : 

In [Chapter 3] Results and Discussion, the methods for each measurement result are described in detail.
All tests were conducted more than five times depending on the type of sample. The average value of the results excluding the maximum and minimum values is shown. The shear strength of the SUS specimens was measured in accordance with ASTM D1002 standards. Apply one drop of the prepared binder to the end of the SUS specimen (width 25.4 mm, height 12.7 mm). Using another SUS specimen, overlap to the same size and confirm that curing occurs at room temperature. Curing is completed, measurement is performed using UTM. The shear strength according to the UA content was measured, as shown in Figure 3. 

The prepared binder for UV-vis measurement was formed to a thickness of 175 μm on a glass slide us-ing a Baker applicator. As with other physical property tests, measurements were com-pleted more than five times. 

The curing process of the synthesized binder was performed in a mold manufactured according to the ASTM D638 standard, and is shown in Figure 5. In this study, a Teflon mold was used to produce dog-bone specimens under ASTM D638 Type IV conditions because a release agent spraying process is necessary to separate the manufactured specimens when performing binder curing in an iron mold. Figure 6 presents the tensile strength of the polymer binder according to the UA content.

7. Fig. 3 could be more readable.
Our Response : 

In response to your opinion, we have added more details to Figure 3.

Curing is completed, measurement is performed using UTM. The shear strength accord-ing to the UA content was measured, as shown in Figure 3. UA synthesized using three different types of polyols was blended with PMMA at 5–10 wt% it showed the highest value and decreased gradually. In this experiment, it was possible to confirm the critical content of UA blended with PMMA. The polymers were physically entangled, resulting in a network structure and high shear strength. It was conducted according to the difference in the polyol type and UA content. The mixture of 5 wt% of UA using PPG as a polyol showed the highest value of 14.6 MPa, as shown in Figure 3 and Table 3. The methyl groups of the main chain were not packed in a linear structure and had a free volume. Therefore, PPG-UA blended with PMMA because of its good compatibility. Compared to PTMG-UA, in the case of PPG-UA, relatively high results were obtained due to the short molecular unit chain length, that is, many acrylate functional groups. As a result, in order to obtain PMMA-based polymer binders with high mechanical properties, optimal design of the appropriate molecular chain structure length is required.

8. No specific conclusions are given. This is probably due to the lack of a clearly stated research goal
Our Response : 

In response to your opinion, additional research conclusion was written in [Chapter 4] conclusion.

In this study, a room-temperature curing type binder series was designed as the most important element technology in externally stimulated polymer binder materials, and the limitations of commercialization were overcome by improving the low mechanical prop-erties of PMMA-based binders. In the design of a binder composed of a blend of PMMA and UA, UA using various types of polyols (PPG, PEG, PTMG) was synthesized to have the same or similar molecular weight to compare its performance evaluation. The pro-posed UA series was synthesized to have a molecular weight of 5–8000. Low molecular weights of 2–3000 showed very low mechanical properties for application to binders, and high molecular weights of 10000 or more were difficult to apply due to processing prob-lems. The optimal mixing ratio was suggested through various content splits so that the designed binder shows excellent mechanical properties. The prepared polymer binder showed improvement in transmittance, shear strength, and tensile strength at UA 5-10 wt%, and an improved polymer binder could be made with structural effects using PPG-UA. PMMA-based binders blended with UA using other types of polyols (PEG, PTMG) also showed slightly improved mechanical properties. In this study, the designed PMMA/UA binder can be cured at room temperature without external energy (light, heat, etc.) due to the heat generated from the reaction with the initiator (BPO) and catalyst (PTE). Binder design that can be cured at room temperature is the most important element tech-nology in externally stimulated polymer binders. In addition, in this study, to overcome the mechanical property limitations of PMMA-based binders, UA was designed using various polyols and improved physical properties were confirmed. The proposed PMMA/UA binder material could be a promising candidate for the future road marking polymer binders.

Reviewer 2 Report

Comments and Suggestions for Authors

1) Abstract: "...adhesive strength... were evaluated...". In fact, adhesive strength refers to a polymer used as an adhesive - which was not subject of the paper. Only the tensile strength was tested.

2) Introduction: "PMMA is used mainly as a polymer for paint but the physical properties are unsuitable when used independently." This is not correct - one of the largest industrial applications of PMMA is artificial glas (see PLEXIGLAS from Roehm) - with excellent mechanical properties.

3) Introduction UAs are of great interest in the 3D printing industry. Many patents have been filed (i.e. from Carbon). Several reviews deal with these materials and should be mentioned, i.e. https://doi.org/10.1080/03602559.2017.1332764

3) You should consider to include pure PMMA as reference for your mechanical properties.

4) Materials and Methods: You need to include how you prepared the cured specimen - it is not sufficient to mention ASTM D 638 in the Results and Discussion. Was a mold release used?

Comments on the Quality of English Language

weak paper - but can be improved.

Author Response

Response to Reviewer 2 Comments

1. "...adhesive strength... were evaluated...". In fact, adhesive strength refers to a polymer used as an adhesive - which was not subject of the paper. Only the tensile strength was tested.
Our Response : 

In response to your comments, I have revised “adhesive” to “shear”.

The content of the blended UA was quantified using ultraviolet–visible spectroscopy at a wave-length of 370 nm based on 5, 10, 15, and 20 wt%, and the shear strength and tensile strength were evaluated using specimens in a suitable mode. The ratio for producing the polymer binder was optimized. The mechanical properties of the polymer binder with 5–10 wt% UA were improved in all series.

2. Introduction: "PMMA is used mainly as a polymer for paint but the physical properties are unsuitable when used independently." This is not correct - one of the largest industrial applications of PMMA is artificial glas (see PLEXIGLAS from Roehm) - with excellent mechanical properties.
Our Response : 

I Agree that PMMA has excellent mechanical properties. I discussed the case where PMMA is used as a paint on road surfaces. In response to your comments, This part has been modified.

A photo stimulated binder that exhibits excellent luminous properties requires the fol-lowing mechanical properties - transmittance that does not reduce the luminous proper-ties of the phosphorescent paint, adhesion to the road surface, weather resistance to with-stand weather changes and toughness to withstand external shocks [9-23]. Therefore, polymethyl methacrylate (PMMA) is a representative polymer used in paint manufactur-ing, but it has the disadvantage of inappropriate physical properties when used inde-pendently. To solve these problems, this study used PMMA and a composite series added with various acrylates. Among numerous acrylates, UA has excellent mechanical proper-ties such as weather resistance, abrasion resistance, and alkali resistance, and structurally exhibits a network or linear structure depending on the type of polyol (diol, triol, etc.). Urethane has the advantage of varying mechanical properties (impact resistance, friction resistance, wear resistance, etc.) depending on the type of polyol and isocyanate. 

3. Introduction UAs are of great interest in the 3D printing industry. Many patents have been filed (i.e. from Carbon). Several reviews deal with these materials and should be mentioned, i.e. https://doi.org/10.1080/03602559.2017.1332764.
Our Response : 

Thank you for your suggestion. In response to your comments, this reference has been added in [Chapter 1] Introduction.

Urethane prepolymers were synthesized using chain structured HDI to maximize the optical properties MDI and TDI with phenyl groups were not used because of yellowing phenomenon. As the acrylate, 2-HEMA, the most widely applied industrially, was used to react with the NCO group at the end of the urethane prepolymer [33-39].
References [33]
Maurya, S.D.; Kurmvanshi, S.K.; Mohanty, S.; Nayak, S.K. A Review on Acrylate-Terminated Urethane Oligomers and Polymers: Synthesis and Applications. Polym. Plast. Technol. Eng. 2018, 57, 7, 625-656. https://doi.org/10.1080/03602559.2017.1332764

4. You should consider to include pure PMMA as reference for your mechanical properties

In response to your opinion, The mechanical properties of urethane acrylate mixed with PMMA were described in Figure 3/4/6. Mechanical properties using pure PMMA were also reported.

PMMA is a representative polymer with high transmittance, and showed similar or improved transmittance at 370 nm when 5 wt% of PPG/PEG/PTMG-UA was used. PPG-UA measured the highest light transmittance at 89.5%, as shown in Table 4.

Figure 6 presents the tensile strength of the polymer binder according to the UA content. The prepared specimen was measured at a speed of 10 mm/min using UTM. In the case of the polymer binder cured with the PMMA resin, the stress was 7.1 MPa, whereas the binder containing 5 wt% PPG-UA showed a stress of 9.7 MPa. PMMA/PEG-UA and PMMA/PTMG-UA showed 5.2 MPa and 3.5 MPa, respectively, at a 10 wt% content. Table 5 lists the stress values for each content of the UA series. PMMA/PEG-UA was expected to show high tensile strength be-cause of the high intermolecular attraction through hydrogen bonding, but the miscibility between PEG-UA and PMMA was poor. PMMA/PTMG-UA affects the strain rather than stress because the chain length of PTMG-based UA is long. The polymer binder prepared with 5 wt% of PMMA/PPG-UA showed approximately 30% improvement in the mechan-ical properties compared to the polymer binder made using PMMA resin.

5. Materials and Methods: You need to include how you prepared the cured specimen - it is not sufficient to mention ASTM D 638 in the Results and Discussion. Was a mold release used?
Our Response : 

In response to your comments, additional information has been described.

The curing process of the synthesized binder was performed in a mold manufactured according to the ASTM D638 standard, and is shown in Figure 5. In this study, a Teflon mold was used to produce dog-bone specimens under ASTM D638 Type IV conditions because a release agent spraying process is necessary to separate the manufactured specimens when performing binder curing in an iron mold. Figure 6 presents the tensile strength of the polymer binder according to the UA content. The prepared specimen was measured at a speed of 10 mm/min using UTM. In the case of the polymer binder cured with the PMMA resin, the stress was 7.1 MPa, whereas the binder containing 5 wt% PPG-UA showed a stress of 9.7 MPa. PMMA/PEG-UA and PMMA/PTMG-UA showed 5.2 MPa and 3.5 MPa, respectively, at a 10 wt% content. Table 5 lists the stress values for each content of the UA series. PMMA/PEG-UA was expected to show high tensile strength be-cause of the high intermolecular attraction through hydrogen bonding, but the miscibility between PEG-UA and PMMA was poor. PMMA/PTMG-UA affects the strain rather than stress because the chain length of PTMG-based UA is long. The polymer binder prepared with 5 wt% of PMMA/PPG-UA showed approximately 30% improvement in the mechan-ical properties compared to the polymer binder made using PMMA resin.
